# The Impact of Advance Directive Perspectives on the Completion of Life-Sustaining Treatment Decisions in Patients with Heart Failure: A Prospective Study

**DOI:** 10.3390/jcm10245962

**Published:** 2021-12-19

**Authors:** JinShil Kim, Seongkum Heo, Bong Roung Kim, Soon Yong Suh, Jae Lan Shim, Minjeong An, Mi-Seung Shin

**Affiliations:** 1College of Nursing, Gachon University, Incheon 21936, Korea; kimj503@gachon.ac.kr; 2Georgia Baptist College of Nursing, Mercer University, Atlanta, GA 30341, USA; heo_s@mercer.edu; 3Department of Internal Medicine, Seoul Medical Center, Seoul 02053, Korea; kbsm702@naver.com; 4Division of Cardiology, Department of Internal Medicine, Gil Medical Center, Gachon University College of Medicine, Incheon 21565, Korea; ssy@gilhospital.com; 5College of Nursing, Dongguk University, Gyeongju 38066, Korea; jrshim@dongguk.ac.kr; 6College of Nursing, Chonnam National University, Gwangju 61469, Korea

**Keywords:** heart failure, advance directives, life-sustaining treatment, knowledge, attitudes

## Abstract

Evidence for non-modifiable and modifiable factors associated with the utilization of advance directives (ADs) in heart failure (HF) is lacking. The purpose of this study was to examine baseline-to-3-month changes in knowledge, attitudes, and benefits/barriers regarding ADs and their impact on the completion of life-sustaining treatment (LST) decisions at 3-month follow-up among patients with HF. Prospective, descriptive data on AD knowledge, attitudes, and benefits/barriers and LSTs were obtained at baseline and 3-month follow-up after outpatient visits. Of 64 patients (age, 68.6 years; male, 60.9%; New York Heart Association (NYHA) classes I/II, 70.3%), 53.1% at baseline and 43.8% at 3-month follow-up completed LST decisions. Advanced age (odds ratio (OR) = 0.91, *p* = 0.012) was associated with less likelihood of the completion of LST decisions at 3-month follow-up, while higher education (OR = 1.19, *p* = 0.025) and NYHA class III/IV (OR = 4.81, *p* = 0.049) were associated with more likelihood. In conclusion, advanced age predicted less likelihood of LST decisions at 3 months, while higher education and more functional impairment predicted more likelihood. These results imply that early AD discussion seems feasible in mild symptomatic HF patients with poor knowledge about ADs, considering the non-modifiable and modifiable factors.

## 1. Introduction

Despite therapeutic advances in heart failure (HF), living with HF, which is characterized by chronic progressive illness, still involves high morbidity and mortality, resulting in a substantial global burden of care [1,2]. Under the circumstances of living with the deteriorating course of the illness, patients with HF are faced with a wide spectrum of decision-making for therapeutic and palliative care issues [3,4,5]. Early incorporation of advance care planning (ACP) discussion and/or advance directives (ADs) into the standard care is highly recommended in HF [6] because it can facilitate informed decision-making [4,6,7,8,9]. ACP is an ongoing communication process, which often begins with exploring an individual’s value on future health, evolves to ACP, and may or may not prepare any form of ADs, such as Durable Power of Attorney for Health Care or Physician Orders for Life-Sustaining Treatment [10]. An AD, as part of ACP, is often prepared as a vehicle to empower an individual to plan their future care at end-of-life (EoL), which refers to time prior to imminent death, but of unpredictable duration [11]. ADs include individuals’ decisions about EoL life-sustaining treatments (LSTs), considering their personal values and preferences, and can be revised if their decisional changes occur [5,12,13]. Particularly, an individual can state one’s preferences for future medical decisions about LSTs, such as cardiopulmonary resuscitation on an AD [10].

Despite the clinical benefits of AD documentation, such as less resource utilization near death and more palliative consultation and hospice care referrals [14,15,16], the suboptimal utilization is still of concern [16,17,18,19]. Several non-modifiable and modifiable factors associated with inadequate use of ADs were reported in HF. Non-modifiable factors cannot be controlled or changed, while modifiable factors can be controlled or changed [20]. Thus, modifiable factors can be changed through interventions to increase the utilization of ADs. Known non-modifiable factors associated with more access to palliative care and/or AD completion included older age, female sex, higher socioeconomic status, white race, and/or other medical comorbidity in HF [9,18,19,21]. Some modifiable factors, such as knowledge or attitudes toward ADs, were also investigated and/or reported to be associated with the utilization of such care by patients with cancer and elderly populations in community and clinical settings [22,23,24,25,26]. However, evidence for these non-modifiable and modifiable factors associated with utilization of treatment directives of ADs, particularly life-sustaining treatments (LSTs), in HF is lacking. Further, assessment of these factors by reliable and valid measures in HF or cardiac areas is scarce.

Thus, to address the gap, it is important to examine non-modifiable and modifiable factors using reliable and valid instruments and their effects on the completion of LST decisions on an AD longitudinally. In this study, we aimed to examine (1) the prevalence for and concordance with the preferences for LST decision on the AD questionnaire, and (2) changes in knowledge, attitudes, and benefits/barriers regarding ADs at 3-month- follow-up. We also examined the effects of baseline-to-3-month changes in these modifiable factors on the decisions of LSTs at 3-month follow-up.

## 2. Materials and Methods

### 2.1. Materials and Methods

Using a prospective, descriptive, correlational design, data on knowledge, attitudes, and benefits/barriers regarding ADs, and LST decisions among patients with HF were obtained at baseline and 3 months later for routine outpatient visits. Patients with HF were enrolled when they visited a university-affiliated outpatient clinic for routine care. This study obtained approval from the institutional review board of the Gachon University Gil Medical Center (GBIRB2017-058, approval date: 9 February, 2017). Prior to collecting data, each patient with HF signed an informed consent statement. Baseline data were collected from August 2017 to May 2018 and follow-up data were collected from November 2017 to July 2018 approximately 3 months later.

A nurse coordinator who was an expert in hospice and palliative care conducted face-to-face interviews for data collection. She assisted patients with HF to complete the survey questionnaires. Three months later, patients completed the same questionnaires, with the exception of demographic and clinical characteristics, with the coordinator’s assistance. Demographic and clinical characteristics were assessed only at baseline. If patients declined the initial invitation or follow-up participation, reasons for declining were pursued if provided.

### 2.2. Participants

Patients were eligible to participate in this study if they met the following criteria: (a) age of ≥21 years, (b) a diagnosis of HF, (c) under optimal medical therapy for at least 6 months, including beta-blockers, angiotensin-converting enzyme inhibitors, angiotensin receptor blockers, statins, and diuretics, and (d) intact cognitive function to engage in AD-related communication. Patients were not able to participate if they met one of the following criteria: (1) end-stage HF which qualifies for palliative and/or hospice care referral, (2) having a terminal comorbidity, such as cancer, acquired immune deficiency syndrome, chronic obstructive pulmonary disease, or chronic liver cirrhosis which are common terminal conditions of AD utilization [27], or (3) having a documented neurocognitive or mental disorder, such as dementia/Alzheimer’s disease or schizophrenia. Cognitive function of each participant was determined by a cardiologist who had an established relationship with the participant. Exclusion criteria were determined by medical record review.

### 2.3. Measures

#### 2.3.1. Decisions of LSTs

Decisions of LSTs were assessed with the Korean-Advance Directive (K-AD) questionnaire [28] at baseline and at 3-months follow up. The K-AD consists of value statement, four LSTs (cardiopulmonary resuscitation (CPR), ventilation support, hemodialysis, and hospice care), and proxy designation. Patients were grouped into two categories based on the responses to the four LSTs. Patients who completed the section on LST decisions were categorized as completers, and those who did not complete it as non-completers. The feasibility of the K-AD questionnaire was reported from both public and clinical settings [28,29].

#### 2.3.2. Knowledge

Knowledge about ADs was assessed using the Knowledge Scale of the Advance Care Planning Survey (23 items) [22,30] at baseline and at 3-month follow up. This scale consists of living will (five items), health care proxy (six items), and LSTs, such as CPR mechanical ventilation and hydration/nutritional support. The possible scores range from 0 to 23 for the total scale, and from 0 to 11 and from 0 to 12 for the living will and proxy and LSTs subscales, respectively. Higher scores indicate greater AD knowledge [30]. The Cronbach’s alphas for the 11-item and 12-item subscales were 0.91 [22] and 0.87, respectively [30].

#### 2.3.3. Attitudes

AD Attitudes were assessed using the Advance Directive Attitude Survey (16 items) [30] at baseline and at 3-month follow up. Possible scores range from 16 to 64; higher scores indicating more positive attitudes toward ADs [30]. The reliability of both the original and the Korean-translated version was acceptable [30,31,32].

#### 2.3.4. Barriers/ Benefits

Perceived barriers and benefits were assessed using the Perceived Barriers and Benefits Scales [22] at baseline and at 3-month follow up. Each scale consisted of nine and seven items, respectively. The possible scores range from 9 to 63 for the Barriers scale and from 7 to 49 for the Benefits scale; higher scores indicating more perceived barriers to and benefits for the ACP. Both scales showed adequate reliability, with Cronbach’s alphas of 0.91 and 0.92, respectively [22].

#### 2.3.5. Demographic and Clinical Characteristics

Demographic characteristics, including age, gender, marital status, and educational level, were obtained using a standard form at baseline. Medical records were reviewed to extract clinical data on left ventricular ejection fraction, medical history, and prescribed medication. Comorbid conditions were also assessed using the Charlson Comorbidity Index, and higher scores indicate more comorbidities [33]. A cardiologist assessed the participants’ HF severity using the NYHA classes.

### 2.4. Statistical Analysis

Data analyses were performed using the Statistical Package for the Social Sciences version 25.0 (IBM Corp., Armonk, NY, USA) [34]. Two-tailed tests with significance level <0.05 were used. Chi-square tests were conducted to examine baseline-to-3-month changes in the LST decisions. Cohen’s kappa coefficients were also computed to examine the extent of baseline-to 3-month concordance in the decisions. Paired *t*-tests and two-way repeated measures analysis of variance were used to examine baseline to 3-month changes (values at 3-month follow-up—values at baseline) in the modifiable variables of knowledge, attitudes, and barriers/benefits regarding ADs between the completers and non-completers of LST decisions. Lastly, multiple logistic regression analysis was used to examine if changes in these modifiable factors from baseline to 3-month follow-up predicted the decisions of LSTs at 3-month follow-up. In the model, all the modifiable factors and also modifiable and non-modifiable demographic and clinical factors (i.e., age, education, and NYHA classification) were entered into the model simultaneously (Enter method).

## 3. Results

A research coordinator invited 71 eligible patients to participate in the study based on the screening results of the cardiologists at the hospital. A final sample comprised 64 patients with HF because seven of the initial 71 patients who completed the baseline questionnaires did not complete the follow-ups. Reasons for incompletion of the follow-ups were missing the appointment (*n* = 4), discomfort or reluctance (*n* = 2), and condition deterioration (*n* = 1). The mean age was 68.6 years (±12.5) (Table 1). More than half (60.9%) were male, and 59.4% were married. The mean education level was 8.4 (±4.6) years. Most patients with HF (70.3%) had NYHA funcitonal class 1 or II. Other sample characteristics are presented in Table 1.

### 3.1. Baseline and 3-Month Decisions about Life-Sustainng Treatments

Approximately half of patients at baseline (*n* = 34, 53.1%) and at 3-month follow-up (*n* = 28, 43.8%) responded to the K-AD LSTs. At baseline, 38.2% of 34 patients who reported preference/no preference preferred CPR, while 17.9% of 28 patients preferred it at 3-month follow-up (*p* = 0.249) (no table). At baseline, 23.5% patients preferred ventilation support, while 10.7% patients preferred it at 3-month follow-up (*p* = 0.195). For hemodialysis, 26.5% at baseline and 3.6% at 3-month follow-up preferred it (*p* = 0.100). For hospice care, 67.6% at baseline and 71.4% at 3-month follow-up preferred it (*p* = 0.356). According to Fisher’s exact tests (*n* = 20 who responded at both time points), none of the changes in the preferences was significant. However, the mean kappa coefficient for ventilation support was .44 (*p* = 0.047), indicating moderate agreement in the preferences between at baseline and at 3-month follow-up (85% no preference) (Table 2). The mean kappa coefficient for hemodialysis was .64 (*p* = 0.002), indicating substantial agreement between both time points (90% no preference).

### 3.2. Baseline-to 3-Month Changes in Modifiable Factors between Completers and Non-Completers of Life-Sustaining Treatments at Baseline and at 3 Month-Follow-Up

Changes in modifiable factors, including knowledge, attitudes, barriers, and benefits, between patients who completed (completers) or did not complete decisions about LSTs (non-completers) from baseline to 3-month follow-up are presented in Table 3. In knowledge, there were significant group (*p* < 0.001), time (*p* = 0.039), and interaction effects (*p* = 0.006). Completers at baseline and also at 3-month follow-up had a higher level of knowledge at baseline and at 3-month follow-up than non-completers. Knowledge in completers at baseline and at 3-month follow-up increased less than non-completers over time. In attitudes, there were also significant group (*p* < 0.001), time (*p* < 0.001), and interaction effects (*p* < 0.001). Completers at baseline and at 3-month follow-up compared with non-completers had slightly more positive attitudes at baseline and 3-month follow-up. Attitudes in both completers and non-completers increased over time, but the increases were slightly higher in completers than non-completers.

In barriers and benefits, there were only significant group effects. Completers at baseline perceived similar or a little bit less barriers than non-completers, while completers at 3-month follow-up perceived similar or more barriers than non-completers (*p* < 0.001). Completers at baseline and at 3-month follow-up perceived similar or more benefits than non-completers (*p* < 0.001).

### 3.3. The Impact of Baseline-to-3-Month Changes in Knowledge, Attitudes, and Barriers/Benefits on the Completion of the Life-Sustaining Treatment Decisions at 3 Month Follow-Up

First, multivariable logistic analysis was conducted to examine the impact of knowledge, attitudes, barriers, and benefits regarding ADs on the completion of LST decisions at baseline (Table 4). Controlling for age, education, and NYHA functional class, knowledge, attitudes, and barriers/benefits at baseline as independent variables were entered to the model. Only education level was identified as a significant predictor affecting the completion of LST decisions at baseline (OR = 1.242, *p* = 0.011, and 95% confidence interval [CI] = 1.050, 1.470). The higher the level of education, the higher the completion of the LST decisions.

Second, to examine the impact of baseline-to-3-month changes in modifiable factors, including knowledge, attitudes, and barriers/benefits, on the completion of LST decisions, another multivariable logistic analysis with Enter method was conducted. Age, education, and NYHA functional class were entered as covariates and changes in the modifiable factors from baseline to 3-month follow-up as independent variables to the model simultaneously. Age, education, and NYHA functional class predicted the completion of LST decisions at 3-month follow-up, while changes in AD knowledge, attitudes, barriers, and benefits did not (Table 4). Advanced age (OR = 0.914, *p* = 0.012, and 95% CI = 0.853, 0.980) was associated with the less likelihood of the completing decisions about LSTs at 3-month follow-up, while higher education level (OR = 1.193, *p* = 0.025, and 95% CI = 1.022, 1.392) and NYHA functional class III (OR = 4.810, *p* = 0.049, and 95% CI = 1.009, 22.939) were associated with the more likelihood of completing decisions about LSTs at 3-month follow-up.

## 4. Discussion

To the best of our knowledge, this study is the first to investigate baseline-to-3-month prevalence for and concordance with LST decisions, and baseline-to-3-month changes in AD-related modifiable factors, such as knowledge, attitudes, and perceived benefits/barriers in patients with HF and their impact on LST decisions among patients with HF. The findings show that patients’ AD knowledge and attitudes were improved over time without an intervention. The improvements might be related to the baseline explanations about the study purpose and the survey questionnaires. Despite the significant improvements in AD variables, changes in AD knowledge, attitudes, barriers, and benefits were not associated with the likelihood of the completion of LST decisions at 3-month follow-up. These findings might be due to relatively small improvements in the variables because of no an active intervention. Thus, further studies are needed to evaluate if interventions improve modifiable variables, and, in turn, improve completion of LST decisions. On the other hand, advanced age was associated with less likelihood of the completion of LST decisions at 3-month follow-up, while higher education level and poor functional status at baseline were associated with more likelihood.

The utilization of ADs in HF has been recommended and integrated into routine practice at the outpatient and the primary care settings in western countries [18,19]. In the United States, the completion rate of ADs in community dwelling patients with HF was 41.0% [19]. In an intervention study conducted in the United States, the rates in patients with HF after hospital discharge were 27.8% at baseline and 47.0% at the 3-month follow-up [35]. However, in East Asian countries, attention to ACP and/or written ADs are becoming more prevalent, thus, the support for such care will be critical for the future plans of care [36]. In South Korea, ADs for non-malignant contexts, particularly HF, have received less attention than those in malignancy contexts. Therefore, patients with HF are still unaware of and reluctant to discuss ADs [29]. However, it is important to utilize ADs in patients with HF because the progress and mortality rates in patients with HF are worse than those in cancer [37,38]. Recently, cardiovascular experts in Korea reached a consensus on the needs for utilization of ADs for patients with advanced HF [8]. In this study, the completion rates of ADs at baseline and at 3-month follow-up were 53.1% and 43.8%, respectively, which were similar or higher than those in the United States. Considering that this was not an intervention study, the completion rates in this study are promising. Thus, introduction of ADs for patients with HF would be beneficial to increase the completion rates.

In this initial address of HF patients’ preferences for LSTs , there were no significant statistical changes from baseline-to-3-month follow-up (CPR, 38.2–17.9%; ventilation support, 23.5–10.7%; hemodialysis, 26.5–3.6%; hospice care, 67.6–71.4%; all ps > 0.05). However, patients kept moderate to substantial agreement in the preferences to ventilation support and hemodialysis. These findings imply that further studies are needed to examine changes in LST decisions and factors affecting the changes. Compared to patients with HF, minor differences were noted in the proportion of EoL treatment preferences in other Korean populations, with 20.5–24% of patients with cancer receiving aggressive treatments at the EoL [39] and community-dwelling elderly individuals [28]. However, hospice care preference was higher in those with cancer (79.5%) [39] and lower in elderly people living in the community (56.4%) [28] than that in patients with HF in this study. Even though the majority of patients in different populations preferred hospice care, a considerable portion of patients still preferred aggressive treatment options. Early and ongoing AD discussion facilitates informed decision-making for future EoL care that may reduce unnecessary and unwanted aggressive treatment and patient and family burdens. Previously, ACP and/or decisional aid interventions for patients with HF have proven that promoting patient–physician communication and facilitating their planning for future care were beneficial to prepare ADs with preferred EoL treatments [9,19,40,41]. Patients with HF who possessed ADs were less likely to receive futile treatments at the EoL and more likely receive EoL care that abides by their values and preferences indicated on the document [19,42,43]. Based on results from the current and past studies, it seems feasible to administer palliative discussion and a written AD among patients with HF early in the course of the illness, while decisional aid interventions are warranted for their informed decisions about future EoL care and increase preparedness of ADs.

In patients with HF, modifiable factors, such as AD knowledge and attitudes, and barriers/benefits have been rarely addressed, while a few studies of patients with cancer and elderly populations in the community and clinical settings investigated those factors and the associations with the utilization of ADs or AD care [22,23,24,25,26]. In this study, knowledge about ADs was substantially poor, and knowledge and attitudes toward ADs significantly improved over time. In addition, there were significant group effects in knowledge, attitudes, barriers, and benefits between completers and non-completers of LST decisions. The completers had similar or higher levels of knowledge and perceived benefits, and more positive attitudes than the non-completers. However, knowledge level was more improved in the non-completers than the completers over time.

We further examined the effects of these modifiable and non-modifiable factors on the 3-month completion of LST decisions. While the baseline completion of LST decisions was more likely to increase with higher education only, the 3-month completion was more likely to decrease with advanced age and increase with higher education. Available evidence from the current and previous studies regarding demographic and clinical factors associated with attitudes toward or access to/utilization of ADs or hospice/palliative care in HF were mixed. Advanced age, female, white race, and higher socioeconomic status of patients with HF were significant demographic characteristics associated with more AD documentation [18]; advanced age and other comorbidities, such as malignancy and renal dysfunction, increased AD completion [19]. Advanced age was more likely associated with utilization of ADs or palliative and hospice care in previous studies, while advanced age in this study was less likely associated with completion of LST decisions on the AD questionnaire. In addition, more education and attention on the decisions about LSTs is needed for low-educated people. None of the modifiable factors, but one modifiable clinical factor associated with completion of LST decisions was NYHA functional class. More severe functional impairment was associated with an increased likelihood of completing the LST decisions. Even though the changes in these variables were not associated with the likelihood of the completion of the LST decisions at 3-month follow-up, the findings of this study imply that the completers had some knowledge about ADs at baseline, and their knowledge level was not changed much over time because this was not an intervention study. In contrast, the non-completers compared with the completers had lower level of knowledge at baseline, and their knowledge level increased more than that in the completers may be due to explanations of ADs during the consent and data collection processes. Thus, the overall findings of this study imply that knowledge and attitudes may be changed even with some explanations about ADs.

Some possible reasons for no associations between changes in those modifiable factors and the completion of the LST decisions at 3-month follow-up may be due to a small sample size, and also no provision of any intervention in this study, which might lead to relatively small changes in the variables. One more possible reason may be a knowledge deficit about ADs (1.39 through 2.70 out of a possible maximum score 23), which can lead to difficulty in making EoL LST decisions and the changes. In a prior study [35], information received about ADs (61.1–94.4%, *p* = 0.001) and completion of AD (27.8–47.7%, *p* = 0.016) were significantly improved after palliative consultation for symptomatic patients with HF (NYHA classes II/III). Thus, completion of ADs can be improved by interventions, including AD education and consultation. Further studies are needed to examine modifiable and non-modifiable factors predicting completion of LST decisions on an AD using larger samples and prospective study designs. Further studies are also needed to develop interventions that can be applied to different populations with different conditions and stages to test the effects on those modifiable factors and completion of future LST decisions on an AD.

### Limitations

One limitation of this study was a small sample. Based on a rule of a minimum of 10 occurrences per a predictor variable [44], at least 70 completers of advance directive treatments were required for this study. Although the study was not an intervention study, the comprehensive questions about ADs might have some intervention effects, leading to increase in knowledge of both the completers and the non-completers and from baseline to 3-month follow-up. Thus, caution is needed to make any conclusive interpretation of the results of this study. The validation of the results needs to be warranted in a larger sample. Further, a homogeneous sample characteristic, which mostly included asymptomatic and mild symptomatic HF patients, limited the generalization of study results.

## 5. Conclusions

Knowledge about ADs was substantially poor. Although knowledge and attitudes were significantly improved over time, the changes were not associated with the likelihood of the completion of LST decisions at 3-month follow-up. Advanced age was associated with the less likelihood of the completion of LST decisions at 3-month follow-up, while higher education level and more functional impairment were associated with an increased likelihood. Early AD discussion needs to be started in patients with HF, especially for patients who are older, have lower education level, and good functional status to facilitate informed decision-making for future EoL care. In patients with HF, a 3-month follow-up visit could be an appropriate time for ACP discussion.

These findings imply that ADs need to be introduced to patients with HF, especially to those who are older, have lower level of education, and have relatively good functional status, and further, to those with acute deteriorations. Advanced age was also a risk factor for less likelihood of the completion of LST decisions. Thus, it is important for healthcare providers to facilitate the culture for open communication between them and their patients regarding ADs, including LST decisions. Patients with HF need to be encouraged to communicate with their healthcare providers regarding their future care comfortably and much earlier to increase their awareness of ADs and involvement in decision-making for their own future care. Future research is also warranted to investigate whether palliative care interventions in the standard care of HF patients actually improve these modifiable factors of knowledge, attitudes, barriers, and benefits, and, in turn, completion of LST decisions on an AD and provision of EoL care that aligns with the patient’s preferences.

## Figures and Tables

**Table 1 jcm-10-05962-t001:** Baseline demographic and clinical characteristics of the patients with heart failure (*n* = 64).

Variables		Frequency (%)	Mean ± SD	Range
Age (years)			68.6 ± 12.5	37–89
Sex	Male	39 (60.9)		
Marital status	Married	38 (59.4)		
Education (years) *			8.4 ± 4.6	0–16
CCI			2.6 ± 1.9	1–10
Heart failure duration (months)			60.5 ± 59.0	6–298
Left ventricular ejection fraction, %			35.8 ± 8.8	17.0–58.0
NYHA classes **	I	8 (12.5)		
	II	37 (57.8)		
	III	19 (29.7)		
	IV	0 (0.0)		
Etiology	ICMP	36 (56.3)		
	DCMP	15 (23.4)		
	HTN	4 (6.3)		
	VHD	4 (6.3)		
	AFib	2 (3.1)		
	Alcoholic	3 (4.7)		
Medication	Beta-blockers	56 (87.5)		
	ACEI	31 (48.4)		
	ARB	17 (26.6)		
	Statin	40 (62.5)		
	Diuretics	33 (51.6)		

Abbreviation: SD, standard deviation; CCI, Charlson comorbidity index; NYHA, New York Heart Association; ICMP, ischemic cardiomyopathy; DCMP, dilated cardiomyopathy; HTN, hypertension; VHD, valvular heart disease; AFib, atrial fibrillation; ACEI, angiotensin-converting enzyme inhibitor; ARB, angiotensin receptor blocker. *: Total years of education starting at kindergarten or first grade. **: One missing.

**Table 2 jcm-10-05962-t002:** Baseline-to-3-month concordance of the preferences for life-sustaining treatments.

Preference for Life-Sustainng Treatments				*p* *	*κ* (*p* ^#^)
Baseline (*n* = 34)		3-Month (*n* = 28)			
		Yes *n* (%)	No *n* (%)		
Cardiopulmonary	Yes, *n* (%)	2 (10.0)	3 (15.0)	0.249	0.29 (0.197)
Resuscitation	No, *n* (%)	2 (10.0)	13 (65.0)		
Ventilator support	Yes, *n* (%)	1 (5.0)	1 (5.0)	0.195	0.44 (0.047)
	No, *n* (%)	1 (5.0)	17 (85.0)		
Hemodialysis	Yes, *n* (%)	1 (5.0)	1 (5.0)	0.100	0.64 (0.002)
	No, *n* (%)	0 (0.0)	18 (90.0)		
Hospice	Yes, *n* (%)	9 (45.0)	3 (15.0)	0.356	0.26 (0.251)
	No, *n* (%)	4 (20.0)	4 (20.0)		

* Fisher’s exact test. ^#^ Applying approximation method. *κ* (kappa coefficient) was used to interpret the Kappa result as follows: values ≤ 0 as indicating no agreement and 0.01–0.20 as none to slight, 0.21–0.40 as fair, 0.41– 0.60 as moderate, 0.61–0.80 as substantial, and 0.81–1.00 as almost perfect agreement. Note. Fisher’s exact test and kappa coefficients were computed based on the number of patients with heart failure who both completed the baseline and 3-month life-sustaining treatment decisions.

**Table 3 jcm-10-05962-t003:** Relationships of changes in knowledge about advance directives, attitudes, and barriers/benefits with changes in completion of the decisions about life-sustaining treatments from baseline to 3-month follow-up.

AD Variables		Completion of Decisions about Life-Sustaining Treatments				Group		Time		Interaction	
		Baseline		3 Month Follow-Up		F	*p* Value	F	*p* Value	F	*p* Value
		Completed (*n* = 34)	Not Completed (*n* = 30)	Completed (*n* = 28)	Not Completed (*n* = 36)						
Knowledge	Baseline	2.12 ± 2.25	1.60 ± 1.45	2.50 ± 2.32	1.39 ± 1.40	47.23	<0.001	4.43	0.039	7.95	0.006
	3-month follow-up	2.59 ± 2.58	2.27 ± 2.16	2.79 ± 2.33	2.17 ± 2.42						
Attitudes	Baseline	46.38 ± 7.36	45.53 ± 7.04	47.36 ± 7.88	44.92 ± 6.47	3289.42	<0.001	22.77	<0.001	24.70	<0.001
	3-month follow-up	51.15 ± 9.12	49.87 ± 7.06	52.14 ± 8.45	49.30 ± 7.86						
Barriers	Baseline	35.32 ± 15.33	34.57 ± 14.50	33.04 ± 16.45	36.47 ± 13.49	315.17	<0.001	3.69	0.059	3.38	0.071
	3-month follow-up	34.06 ± 18.76	26.70 ± 19.73	31.89 ± 21.02	29.61 ± 18.33						
Benefits	Baseline	36.29 ± 12.43	35.70 ± 11.49	38.68 ± 12.24	33.94 ± 11.37	1045.40	<0.001	3.26	0.076	3.63	0.061
	3-month follow-up	39.79 ± 11.90	38.80 ± 11.17	41.64 ± 9.20	37.53 ± 12.82						

Abbreviations: AD, advance directives.

**Table 4 jcm-10-05962-t004:** Factors associated with completion of life sustaining treatment decisions at baseline and at 3-month follow-up: Logistic regression.

Wave	Factors	Baseline: Completion of Life-Sustaining Treatment Decisions				
		B	*p* Value	Exp (B)	95% CI	
					Lower	Upper
Baseline	Constants	2.529	0.396	12.545		
	Age	−0.034	0.210	0.967	0.917	1.019
	Education	0.217	0.011	1.242	1.050	1.470
	NYHA class	−0.633	0.335	0.531	0.147	1.922
	Knowledge total	−0.132	0.489	0.877	0.604	1.273
	Attitudes	−0.051	0.273	0.950	0.867	1.041
	Barriers	0.007	0.728	1.007	0.967	1.049
	Benefits	0.009	0.743	1.009	0.954	1.068
	Model summary: chi-square = 15.33, *p* = 0.032, Nagelkerke R^2^ = 0.284					
		3-Month Follow-Up: Completion of Life-Sustaining Treatment Decisions				
Baseline	Constants					
	Age	−0.089	0.012	0.914	0.853	0.980
	Education	0.176	0.025	1.193	1.022	1.392
	NYHA class	1.571	0.049	4.810	1.009	22.939
Changes from baseline to 3-month follow-up	Knowledge total	−0.393	0.056	0.675	0.451	1.009
	Attitudes	−0.032	0.500	1.032	0.941	1.132
	Barriers	0.019	0.365	0.981	0.943	1.022
	Benefits	−0.017	0.475	1.017	0.971	1.066
	Model summary: chi-square = 24.34, *p* = 0.001, Nagelkerke R^2^ = 0.424					

B, unstandardized beta; Exp (B), exponentiation of the B coefficient; CI, confidence interval.

## Data Availability

The data are not publicly available because we did not obtain approval from the IRB for the data availability at that time.

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
