# Peer review of "The Impact of Advance Directive Perspectives on the Completion of Life-Sustaining Treatment Decisions in Patients with Heart Failure: A Prospective Study"

_jcm, 2021, doi:10.3390/jcm10245962_

Round 1

Reviewer 1 Report

The authors report a prospective study asking about the presence of advance directives and barriers and facilitiators to their completion at baseline and after 3 months.

General concept comments

For the measurements, I had a hard time understanding which documents belonged to the baseline questionnaires. Furthermore, it should be made clearer who belonged to the population of 64 eligible patients. In the introduction it is described that the rate of completion is determined by the completion of a section of the K-AD. However, the results start with many missings in the K-AD. For me it remained unclear which section was meant. Basically, I would have liked to have heard something about the non-modifiable factors as well, as well as briefly something about barriers/benefits. Overall, the sample is very small, which limits the quality of the statements considerably. And even though the study is not an intervention study, the comprehensive questions about AD alone are at least a small intervention. They do encourage people to think about it. This should at least be mentioned as a bias in the limitations. Unfortunately, no current literature is cited except for a 2020 citation. The links provided from 2018 unfortunately do not work. Also, I find a 3-year-old access to internet sites that has not been updated too old.

Specific comments 

Line 50 a brief explanation of modifiable and non-modifiable factors would be nice.

Line 60 The paragraph, starting with line 60, belongs in my eyes in the methods section, not in the introduction.

Line 74 How long did the survey phases last both at baseline and after 3 months? And when were they performed?

Line 83 The reasons for declining are mentioned but not reported.

Table 1 The abbreviations do not match (ICM vs. ICMP; DCM vs. DCMP etc.)

Table 2 No units are specified here, which makes interpretation difficult.

Table 3 Knowledge, attitudes, barriers, and benefits are marked with a small *. I can't find anywhere what it means.

Line 238 Why only as long as the health status remains relatively good? Aren't acute deteriorations often a reason to think about AD again?

Line 239 ff. This paragraph seems a bit out of context. At this point, he does not pick up on any of the study's findings. Perhaps the role of palliative care could be better integrated elsewhere? Further, explain why palliative support will be critical. It is not whether AD are predominantly used in the context of malignancy that should be discussed here. rather, the focus is on educating patients about their prognosis. Only against this background can decisions concerning AD be made comprehensively.

Line 266 ACP Abbreviation is not yet introduced

Line 387 f. The provided link does not work – and you were not looking for updates since 2018?

Line 427 f. The provided link does not work

Line 443 f. The provided link does not work – again, no updates since 2018?

Author Response

Dear Reviewers:

On behalf of my co-authors, I would like to thank you and the reviewers for your thoughtful comments and recommendations. We have made the following changes to the manuscript to address your concerns. The revisions are color-coded in RED in the text. Please see the attachment.

Sincerely,

Authors

Reviewer 2 Report

In the whole text, several terms are used sometimes erratic:

  • The proper nomenclature is “Advance directives”, but in the manuscript term “advance treatment directives” is commonly used. The definition of AD is – that is the form of documenting decisions regarding future “care and treatment”.
  • The relationship between ACP and AD and not clearly stated. The whole process should grow from communication on future health, evolving to ACP and being finalized, if appropriate with any form of formal statement (AND, ACP, POLST etc). This is not that visible in the manuscript.
  • ACP / AD can, but does not have to lead to less invasive treatment, choosing quality of life oriented approach (this information is missing in the manuscript), AD can state wish for resuscitation, ventilation or any other intervention.

If older age is the risk factor for not having AD, maybe a suggestion would be to start the culture of speaking about future health much earlier, it does not have to be ACP at once, it is enough to communicate about the future, make the people aware, that they can participate in decision-making etc.

The EoL is often misunderstood as just proceeding the process of active dying. The term in the manuscript is used correctly, but to strengthen the message – including the correct definition of EoL would make the message clearer.  

Abstract:

Row 26 - It is stated, that older age is the factor lowering the probability of having AD,   but the discriminator of older age has been not indicated. Please give what, in your manuscript you use as criterion for older age.

Row 41

The sentence about incorporation of ACP and/or AD is confusing. It would be probably good to express that AD should conclude the person signing it can/should express her or his values and wishes, important for future care and treatment.

Rows 45-49

Similar than above-mentioned: the statement reduces AD to treatment, and somehow confusing term of “treatment directives” is emerging. The outcome of ACP / AD does not have to be limitation, that usual is, but not the precondition.

Row 88

How the cognitive function has been assessed / defined?

Row 94 – the term “mental disorders” is used twice. What exactly means “mental disorders” – does it mean the people having depression has been excluded?

Rows 173 and 174

The sentence seems like not being completed, maybe it is about modifiable factors…  influencing the perceivement or perception of AD? It is hardly to imagine what it should mean “Factors for Advance Directives” The same issue I to seen in the Section “Discussion” Rows 222-223.

Rows 179 – 181

Contains the same information, repeated information.

The discussion all above mentioned remarks – need  to be evaluated as suggested and the English requires some smoothing

Author Response

(The authors gave the same response as above.)

Round 2

Reviewer 1 Report

Dear authors,

thank you very much for your feedback and for reviewing my concerns.

Only a few concerns remain.

Specific comments

 Line 90 I’m sorry, I can see that the 7 cases were excluded because of the missing follow up date. But I can’t find anywhere the reasons for initial declining. They are still not mentioned.

Line 265-7

I would formulate this statement a little more cautiously and more as a recommendation. Ultimately, there can be no obligation for ADs.

Line 416 f

Sorry, the link still does not work. Neither with copy and paste, nor with the direct link.

Line 426 f

Also, this link does not work – maybe it is because of uptodate or due to my network settings??

There are no year numbers when accessing. (Also ref. 6)

I found both references through Google search.
